# Facilitators and barriers for the implementation of a transmural fall-prevention care pathway for older adults in the emergency department

W. M. Charmant[1,2]*, B. A. M. Snoeker[1,2], H. P. J. van Hout[2,3], I. N. Nauta[1], F. Boonstra[1], E. Geleijn[4], C. Veenhof[5,6], P. W. B. Nanayakkara[1,2]

1 Section General Internal Medicine, Department of Internal Medicine, Amsterdam Public Health Research Institute, Amsterdam UMC Location VUmc, Amsterdam, The Netherlands, 2 Amsterdam Public Health Research Institute, Amsterdam, The Netherlands, 3 Departments of General Practice & Medicine for Older Persons, Amsterdam Public Health Research Institute, Amsterdam UMC Location VUmc, Amsterdam, The Netherlands, 4 Department of Rehabilitation Medicine, Amsterdam UMC Location VUmc, Amsterdam, The Netherlands, 5 Department of Rehabilitation, Physical Therapy Science and Sport, Brain Center, University Medical Center Utrecht, Utrecht University, Utrecht, The Netherlands, 6 Expertise Center Healthy Urban Living, Research Group Innovation of Mobility Care, University of Applied Sciences Utrecht, Utrecht, The Netherlands

* w.m.charmant@amsterdamumc.nl

## Abstract

### Background

Older adults at the emergency department (ED) with fall-related injuries are at risk of repeated falls. National guidelines state that the ED is responsible for initiating fall preventive care. A transmural fall-prevention care pathway (TFCP) at the ED can guide patients to tailored interventions. In this study, we investigated the facilitators and barriers for the implementation of the TFCP for older adults following a fall-related ED visit from the perspectives of patients and healthcare professionals.

### Methods

In this qualitative study, we used semi-structured interviews with ten older adults who had a recent ED visit due to a fall. Furthermore, we organised focus groups with 13 healthcare professionals involved in TFCP. Two researchers independently coded the transcripts using inductive thematic analysis.

### Results

We revealed facilitators and barriers on three key themes: 1) Communication, 2) organisation & execution, and 3) personal factors, and thereunder 12 subthemes. Our specific finding were 1a) communication between healthcare professionals and patients, 1b) interprofessional communication and 1c) communication between patients and their family or friends can have both positive and negative impact on the implementation of a TFCP. For the organisation & execution, facilitators and barriers were mentioned for 2a) processes at the ED,

**Data Availability Statement:** In our approved protocol and participant informed consent, we

agreed that interview data would only be shared upon request of individual researchers. Notably, 21 out of 23 participants have agreed for their data to be shared upon request with researchers outside of Amsterdam UMC. We believe that these ethical and legal reasons are sufficient for an exemption request. To complete the statement we provided the chair of the research ethics board that approved the study: Prof. Dr. J.A.M. van der Post metc@amsterdamumc.nl.

**Funding:** Receiving author: B.A.M. Snoeker Grant number: 2008954-20201205 Full name of the funder: Wetenschappelijk College Fysiotherapie van het Koninklijk Nederlands Genootschap voor Fysiotherapie URL: https://www.kngf.nl/article/vak-en-kwaliteit/onderzoek/wcf/wetenschappelijk-college-fysiotherapie Sponsors did not play any role in the study design, data collection and analysis, decision to publish, or preparation of the manuscript.

**Competing interests:** The authors have declared that no competing interests exist.

2b) the fall risk assessment, 2c) patient engagement, 2d) finances, 2e) time, and 2f) responsibilities. Personal factors such as 3a) emotions and behaviour, 3b) knowledge, and 3c) motivation play a crucial role in the success of patient participation. The competence of healthcare professionals in geriatric care facilitate the process of tailoring of care to patients' needs.

## Conclusion

When implementing a TFCP, it is crucial to be aware that facilitators and barriers from the perspective of patients and healthcare professionals exist in the processes of communication, organisation & execution, and personal factors. These factors guide the development of tailored implementation strategies in ED and primary care settings.

## Background

Falls are a leading cause of injury among older adults [1]. In the Netherlands, every five minutes an older adult visits the emergency department (ED) due to fall-related injuries, with a total of 103.000 visits in 2020 [2, 3]. Consequently, injuries from falls result in disabilities, affecting both short-term and long-term physical functioning, participation and mortality, and increase health care costs [1, 4–7].

In a Dutch study, one in five older adults with ED fall-related visits returned to the ED within a month [8]. In the USA, between 12.6–13.6% revisited the ED within 30 days, of which 10.3–17.4% due to a subsequent fall [9, 10]. Recurring fall incidents can decrease mobility, which lowers independence, resulting in a reduced quality of life and an increase in mortality risk [1]. Adequate interventions may prevent subsequent falls and ED returns [11].

As fall risk can be related to multiple domains, it is important to perform a multifactorial fall risk assessment to select personalised interventions such as exercise programs [12]. There are currently few referrals to fall prevention exercise programs after an ED visit. Additionally, individuals rarely sign up for these effective programmes on their own, and adherence rates are low [13–15].

According to the Dutch national guideline for prevention of fall incidents in older adults, healthcare personnel at the ED is responsible for assessing fall risk and initiate actions to prevent future falls [16]. However, the ED is a challenging environment to initiate fall prevention interventions, as the ED is primarily focused on managing the acute care needs of patients as opposed to prevention [17]. A pilot study in Amsterdam UMC investigated the feasibility of implementing a transmural fall-prevention care pathway (TFCP) at the ED [18]. In this TFCP, patients are referred from the ED to community-based physiotherapists for multifactorial fall risk assessments to reduce fall risk. During the pilot study, the researchers found that ED patients' willingness to participate in multifactorial fall prevention programs was high. These promising results provide compelling reason to implement a TFCP as standard care in the ED. When implementing innovations, such as the TFCP, implementation strategies are commonly employed [19]. Implementation strategies are defined as methods or techniques used to enhance the adoption, implementation, and sustainability of the TFCP [19]. To select and develop implementation strategies, it is crucial to gather experiences from patients and healthcare professionals on potential facilitators and barriers that may arise during the implementation of a TFCP [20].

The aim of this study was to identify the facilitators and barriers for the implementation of a TFCP for older adults following a fall-related ED visit from the perspectives of patients and healthcare professionals.

## Methods

### Research design

A qualitative study was performed to investigate the facilitators and barriers from the perspective of patients and healthcare professionals for the implementation of a TFCP from the ED to primary care for older adults. The Medical Ethical Committee of VU Medical Centre approved the study (METC VUmc 2021.0451). The COREQ criteria were used to report the qualitative study (S1 Appendix).

### Theoretical framework

The research paradigm was "constructivism" since the truth of the facilitators and barriers was constructed by and between the researchers and the patients or the healthcare professionals. The reality of the participants was socially and experientially based, explaining the ontology as "relativism". The epistemology was radical "subjectivism"; knowledge of this reality originates from constructs that arose from interactions between researchers and participants. The methodology was a qualitative description, as we tried to answer the "what" research question on facilitators and barriers from the participants' perceptions [21–26].

### Participant recruitment and sampling

Participants were patients and healthcare professionals. The sample size was not predetermined as it depended on data saturation. All patients had experienced an ED visit after a low energetic fall in October 2021. At the time of the fall, patients were 65 years or older and lived (semi) independent. They were able to sign an informed consent form and were willing to participate in an interview.

The participating healthcare professionals were representatives of the professions that are active in a TFCP. The healthcare professionals were regularly involved in care for older adults after a fall. ED nurses and ED doctors were approached via the team manager. Physio- and occupational therapists were experienced with fall risk assessments as they were affiliated with the pilot study. We approached GPs and practice nurses (PNs) via the program manager for elderly care at the local general practitioners cooperative. All were recruited in December 2021.

Patients were recruited via the ED during October 2021 using purposeful sampling for age with the intention of achieving an equal division in age groups. Healthcare professionals were recruited through Amsterdam UMC and partnering community-based settings in Amsterdam, the Netherlands. We approached 22 patients via telephone; five refused to participate because they were not interested or too busy, and seven did not respond. The number of ED nurses, ED doctors, GPs and PNs who declined to participate is unknown for us due to the indirect approach. We approached 12 physio- and occupational therapists via mail, and eight did not respond. All participants provided written consent before the start of the interviews.

### Setting

Patients and healthcare professionals were interviews separately to prevent (future) interference in patient-healthcare professional relationships. The interviews with the patients were

one-on-one and at the patients' homes. Interviews at home ensured comfort and eliminated travel barriers for patients to participate.

The two focus groups with healthcare professionals were online via Microsoft Teams (Microsoft Corporation, Redmond, Washington, United States). In the first focus group, we discussed facilitators and barriers as experienced by the ED nurses, ED doctors and physio-therapists. In the second focus group, we discussed the same subject from the perspectives of GPs and PNs. We used focus groups for the healthcare professionals as this form of group interviews generally provides more interaction than regular group interviews [27, 28]. This interaction is especially relevant for the interprofessional aspects of the TFCP.

### The transmural fall-prevention care pathway

We designed the concept of TFCP for older adults after a fall-related ED visit together with the consortium. The concept was based on results from the pilot study [18]. The TFCP can be divided roughly into four stages. In stage 1, eligible patients are identified by ED nurses and ED doctors at the ED of Amsterdam UMC, location VUmc. Once discharged, patients are visited at home by one of the community-based physiotherapists or occupational therapists in stage 2. The therapists then assess the presence of fall risks. Based on their analysis, the therapists discussed the indicated care with the patient to reduce fall risk using shared decision making [29]. The GP and PN are informed of the activation of the TFCP via the ED discharge letter. In the third stage, the GP and PN receive the results and conclusion of the discussion from the therapist. Patients make use of the indicated care in the fourth stage, as their GPs receive reports of their progress from healthcare professionals, similar to usual care (Fig 1).

### Data collection

Patients and healthcare professionals were asked for their demographic characteristics before the interview. The patient's demographic characteristics included age, sex, living alone or together, experience with fall prevention programmes, the use of district nursing and the number of falls the month after the ED visit and number of falls in the past year prior to the ED visit. The electronic patient files were used to collect the number of patients who returned to the ED of Amsterdam UMC, location VUmc, within six months. The characteristics of the healthcare professionals included age, sex, profession and number of years with experience in the care for older adults.

The interviews were semi-structured using an interview guide with seven themes based on previous literature and brainstorm sessions with the research group and consortium (Table 1, S2-S4 Appendices). During the interviews, we reserved time for participants to start discussions on new themes. The interview guide was tested with patient group representatives to assure patients' perspectives in the interviews. Interviews with patients lasted 45 to 60 minutes. The interviews with the patients were performed by WMC and IN (author initials). The focus groups with the healthcare professionals were performed by WMC, FB, MvB and EG. The first focus group lasted two hours, and the second lasted one hour.

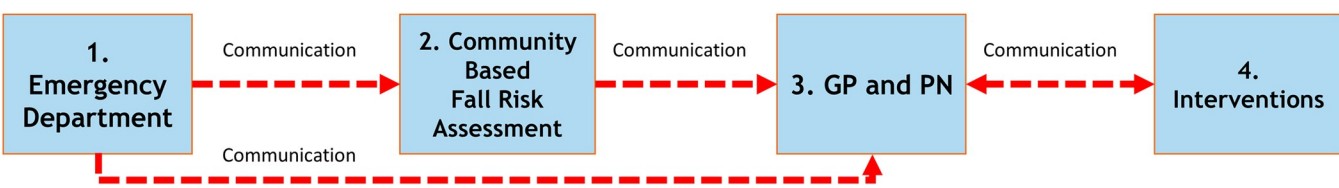

**Fig 1. The transmural fall prevention care pathway.** GP = General practitioner and PN = Practice nurse.

**Table 1. The interview guides.**

| Topic | Patients | Healthcare professionals |
|---|---|---|
| Introduction | The interviewer introduces the TFCP and explains the purpose of the interview. | The interviewer introduces the TFCP and explains the purpose of the interview. |
| Theme 1: Emergency Department | Learning from patient ED experiences and defining optimal approach and information for patients to participate in the TFCP. | Gathering current experiences with target population identification and the CFS. Second, defining the most easy and efficient organisation of patient identification. |
| Theme 2: Communication between ED and Physiotherapist | - | Identifying current communication between the ED and community based physiotherapists and constructing a bilateral approved communication pathway. |
| Theme 3: Frailty Screening | Collecting current experiences of frailty screening and identifying possible barriers. | Learning from current frailty screening experiences and collecting necessities for optimal screening by the physiotherapist. |
| Theme 4: Communication between Physiotherapist and PN/GP | - | Identifying current communication between the community based physiotherapists and PN/GP. Hereafter, constructing a bilateral approved communication pathway. |
| Theme 5: PN/GP | Investigating current PN/GP involvement after an ED visit. | Collecting current experiences of PNs/GPs after an ED visit of their patients. Constructing the effective role of the PN/GP in the TFCP. |
| Theme 6: Communication between PN/GP and the intervention giving healthcare professionals | - | Identifying current communication between PN/GP and the intervention giving healthcare professionals. Hereafter, constructing a bilateral approved communication pathway. |
| Theme 7: The Interventions | Learning current intervention experiences by patients and factors influencing initial participation and continued participation. | Gather current experiences of organisation, roles and availability of interventions. |
| New themes | | |

The interviews were recorded (audio with patients, video with healthcare professionals) and transcribed ad verbatim by WMC, IN and FB. During the interviews and focus groups, WMC, IN, and FB documented field notes to support the flow of the conversation by marking points to revisit as needed and recording notable insights. These notes also served as a reference for observations made during the discussions, allowing verification of whether these themes emerged in the subsequent data coding process. Interim analysis provided targeted questions for participants to engage with ideas from previous interviews. No major new interview themes were identified.

## Personal characteristics of the interviewers

WMC is a male physiotherapist, movement scientist and PhD student. He previously conducted a qualitative study with interviews on the therapeutic alliance in elite athletes [30]. IN and FB were both female physiotherapists, specialising in geriatric physiotherapy. MvB and EG have previous experience with leading focus groups. MvB is a senior advisor and experienced focus group leader [31]. EG is an physiotherapist and care innovator.

There was no relationship between the interviewers and the patients prior to the study commencement. There was a work relationship between the interviewers and the healthcare professionals. Participants were aware that the study was in preparation for the implementation of a TFCP.

## Data analysis

Data analysis started after the first interviews. The transcribed recordings were coded by WMC, IN and FB using inductive thematic analysis in MAXQDA 2020 (VERBI GmbH, Berlin, Germany) [32, 33]. With open coding, important words and groups of words were identified and labelled into categories. During subsequent axial coding, the formed categories are subsumed under other categories into themes. Inductive thematic saturation was achieved in

the patient interviews, as no new themes emerged after seven interviews. To ensure saturation, we continued to ten interviews. The evaluation of inductive thematic saturation was parallel to the data analysis process.

## Quality criteria of qualitative research

Multiple precautions were taken to assure the credibility, transferability, dependability and confirmability of the study [34–36].

Credibility is the trustworthiness and believability of the findings. Individuals with different perspectives were considered and their input was triangulated. Three investigators were involved in coding, analysis, and interpretation decisions, known as investigator triangulation. WMC and IN were involved in the patient interviews, and WMC and FB were involved in the focus groups with the healthcare professionals. All patients were given the opportunity to respond to the results in a mailed member check.

Transferability is the extent to which the results are applicable in similar settings. By speaking to patients and all professions involved in a TFCP, we believe that the results may apply to regions with similar care infrastructures.

The dependability is the reliability of the analytical process. It relies on its consistent adherence to the required standards. Iterative data collection and data analysis were used to collect all possible variability within the results.

The confirmability concerns the neutrality of the results. By adding the background of the interviewers, the researchers provide insight into possible biases and increase confirmability.

## Results

Ten patients and 13 healthcare professionals were interviewed between November 2021 and April 2022. In two interviews with patients, a family member was present. The patients had a mean age of 78 years (SD ± 2.3), and 80% were female. The median number of days since their ED visit was 46.5 (IQR 41.0–52.0). Eighty percent of the patients lived alone in their homes, and three patients used district nursing. One patient had another fall within a month of the first ED visit. Only two patients could remember another fall in the past year. None of the patients had experience with participating in a fall prevention program. The median Clinical Frailty Score was 3 (range 2–5) [37]. Four visits by three patients to the ED of Amsterdam UMC, location VUmc, within six months after the falls in October 2021 were observed in the electronic patient files. The healthcare professionals had a mean age of 46.5 (SD ± 24.2), and 76.9% were female. Three ED nurses, one ED doctor, four physiotherapists, one GP, and four PNs participated in the focus groups. The median years of experience in care with older adults was 35 years (IQR 11–35) (Table 2, S1 and S2 Tables).

Three key themes were identified in the analysis of the data: communication, organisation & execution, and personal factors. Several subthemes were identified within each theme (Fig 2). Illustrative quotes are noted with a transcript ID number and participant characteristics: P = patient; EDN = emergency department nurse; EDD = emergency department doctor; GP = general practitioner. The presented factors are classified as facilitator or barrier based on how the participants portrayed it. In general, all factors are from the participants themselves and not interpretations of the research team. Original Dutch quotes can be found in S5 Appendix.

## 1. Communication

Communication was discussed most often. It was categorised into communication with the patient, the role of family and friends, and interprofessional communication.

**Table 2. Demographic characteristics of the patients and healthcare professionals.**

| Characteristic | Patients (n = 10) |
|---|---|
| Age in years, mean | 78 (SD ± 2.3) |
| Female, n (%) | 8 (80) |
| Days since ED visit, median (IQR) | 46.5 (IQR 41.0–52.0) |
| Patients with a fall in the past month, n (%) | 1 (10) |
| Patients with a fall in the past year prior to ED visit, n (%) | 2 (20) |
| Living alone, n (%) | 8 (80) |
| District nursing, n (%) | 3 (30) |
| Experience with fall prevention programmes, n (%) | 0 (0) |
| Clinical Frailty Score, median (range) | 3 (2–5) |
| Patients with ED-returns within six months, n (%)* | 3 (30) |
| Characteristic | Healthcare Professionals (n = 13) |
| Age in years, mean | 46.5 (SD ± 24.2) |
| Female, n (%) | 10 (77) |
| Profession, n (%) | ED nurse: 3 (23) |
| | ED doctor: 1 (8) |
| | Physiotherapists: 4 (31) |
| | GP: 1 (8) |
| | PN: 4 (31) |
| Experience in care with older adults in years, median | 35 (IQR 11–35) |

\* = Only the returns to the ED of Amsterdam UMC, location VUmc, were collected. SD = standard deviation, ED = emergency department and IQR = interquartile range.

**1.1. Communication between patients and healthcare professionals.** In all stages of the TFCP, communication was considered to be an important aspect for the success of the implementation and execution. Starting at the ED, some patients indicated that they were quite overwhelmed as they were in pain, were in shock, felt lonely or were afraid. These circumstances made it difficult for them to remember information. Patients preferred that the explanation of the TFCP at the ED would be short and information should be available in other sources, such as a pamphlet.

Furthermore, patients mentioned that the explanation should happen with empathy and attention for the patient. This conversation could be with an ED doctor or ED nurse but should not be hierarchical or belittling. Patients stated that a conversation would work best with the healthcare professional they have the best relation with. The ED healthcare professionals agreed that both ED doctors and ED nurses can perform the conversation and that it is possible within the ED setting. Patients indicated that the conversation should be carried out with a positive approach in which the benefits of the TFCP should be explained. Moreover, the content, costs, location and time investment of the TFCP should be clear. The importance of participating can be emphasised by explaining the risks and benefits of participating to the patient. Patients provided some examples of benefits, such as help with staying mobile, staying active in the community and independent living. Labelling the TFCP as the initiative of the hospital would make the TFCP more trustworthy for patients.

Regarding the second stage of the TFCP, patients stated they would appreciate it if therapists would call to make an appointment in which the patient could indicate a preferred time. Rescheduling of appointments by therapists would have demotivating effects. All patients indicated that it is important to discuss the findings of the fall risk assessment and to discuss the next steps together. It would help to explain the content of possible interventions so patients

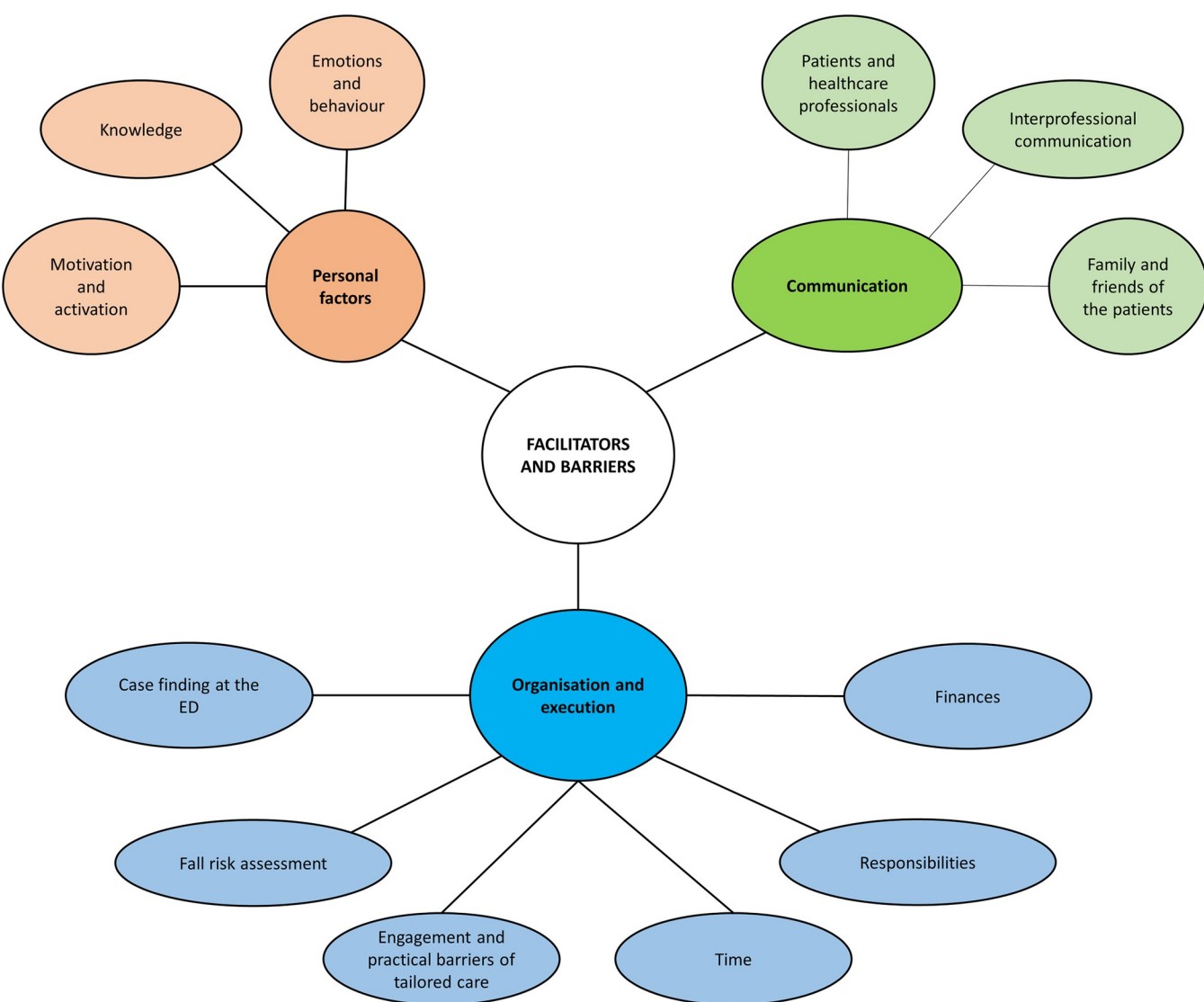

**Fig 2. Facilitators and barriers for the implementation of a transmural fall-prevention care pathway.**

can make thoughtful decisions. The explanation could encourage interest from the patients if they heard something novel about the intervention. Therapists could also present the risks of the current situation to help illustrate the necessity of interventions. Almost all patients stated that receiving a copy of the action plan would motivate them.

In the third and fourth stages of the TFCP, the communication between the patient and GP or PN could be an important facilitator if the patient has a good relationship with the GP or PN. However, the relationship can also be a barrier, as multiple patients stated that they did not have a good relationship with their GP. In some, this originated from past experiences or avoiding the GP to prevent burdening their GP. Related to the workload of the GP, patients themselves indicated that the PN could have an important role within the TFCP. The PNs stated that it should be doable to contact patients after receiving the referral from the therapist. This could stimulate patients to use the indicated care.

In general, patients described the positive effects associated with the availability of a professional. It does not matter whether it is a doctor or nurse as long as the patient feels that he or

she is in "good hands". A conversation with a healthcare provider could motivate a lot if the patient feels genuine attention for their problems and solutions. It could make patients more aware of their possibilities. Moreover, the goal of the TFCP should relate to the functional limitations the patients encounter in activities of daily living. All patients indicated that the TFCP was a great idea, but not everyone felt like they belonged to the target group. Terms such as fall prevention did not feel applicable to all patients.

> WMC: "*Could patients be motivated to participate in indicated care if the GP or PN would contact you after he or she receives the transferal from the therapist?*"
>
> P-9: "*Absolutely, presuming a good relationship between patient and GP.*" (Q1)
>
> P-2: "*Yes, if I would trust GPs again than it would be okay.*" (Q2)

**1.2. The role of family and friends in communication.** During all stages of the TFCP, partners, family, friends and neighbours of the patient could be important facilitators. Both patients and healthcare professionals indicated that it could help when friends or family are present at the explanation of the TFCP at the ED but also to continue participating in interventions. Positive associations with fall prevention by friends or family could be used to motivate patients to try an intervention. Groups within interventions could contribute to patient motivation as well.

> WMC: "*How could the physiotherapist stimulate you to do exercises?*"
>
> P-8: "*If you feel the necessity yourself, then I think you'll do it, but I don't have that feeling right now. [Patient 8 got a brace from the hospital] However, when my son said*: "*If we bike to the garden next week, you'll wear your brace*". *I said*: "*I will*". (Q3)

**1.3. Interprofessional communication.** For the first stage of the TFCP, the GP indicated that it is relevant for the ED to have access to the GP's latest reports to prevent duplication of efforts, as patients might already be in a care pathway and receive fall prevention interventions.

> GP-1: "*We can't, with all respect, implement fall prevention if the ED is unaware of the interventions that have already been initiated by the GP.*" (Q4)

After referring patients from the ED to the GP, ED professionals expressed their concerns about the degree to which the GPs would act on the ED's advice in the discharge letters. As there are no feedback loops to the ED from GPs, they find it hard to assess the effect of their advice. There is no consensus on how the ED might communicate with the community-based physiotherapists to ensure follow-up of the patients. All involved healthcare professionals concluded that the ED should motivate the patient to call the physiotherapists themselves for an appointment. Moreover, the ED should ask the GP in the discharge letter to contact the patient to use the TFCP. A mentioned barrier in the last step of this approach is the time that passes before the GP acts. In general, the discharge letter is sent within 48 hours. The GP deemed it realistic to act within five days. The PNs would like a contact person in the hospital for consultation when necessary.

> WMC: "*What are factors that could improve or hinder the communication between you (GP's office) and the hospital?*"

> PN-1: *"I would like a contact person in the hospital for me as PN, and a call so we could discuss the discharge letter."* (Q5)

The possibilities of direct contact between the ED and the physiotherapists must be further explored during implementation. The approach used within the Transmural Trauma Care Model, another transmural care pathway, was discussed with the healthcare professionals [38]. In this care pathway, hospital based physiotherapists match the patient with one a primary care physiotherapists based on address. ED doctors deemed this approach too time extensive. If possible, the physiotherapists indicated that they would like to receive the following information from the ED: name, address, medical history, ED diagnosis and clinical frailty score. The physiotherapists suggested using 'Zorgdomein', a digital platform for secure communication between healthcare professionals. The ED indicated that web-based programmes such as Point might be more suited for the long term.

> EDN-1: *"For the future I believe a web based programme like Point is more suitable. Point is currently used by our transfer nurses to organise post hospital care. I think that will be easier."*
>
> MvB: *"And how does something like Point work?"*
>
> EDN-1: *"Right now we work with an ambulance planner to order transport, it is sort of the future. If we can simply go to an address, indicate with a few clicks who and what is coming, like a physiotherapist, I think that works the fastest and will be the best plan in the long term."* (Q6)

In the third stage of the TFCP, the transferral of the results from the fall risk assessment to the GP plays a central role. Healthcare professionals stated that the difficulty of the transferral is influenced by the existing professional relationship between the physiotherapist and GP. It will save time if they already know each other and are easily accessible to each other. To facilitate the transferral, it was recommended by the participants that the physiotherapists make a phone appointment with the GP to further elaborate on the findings. As it might be time intensive to make such an appointment happen, the healthcare professionals recommended two ways to contact the GP or PN. The first option was directly via Siilo, a secure instant messaging platform for healthcare professionals. Another option was to ask the GP's secretary to make a note in the GP's agenda. GPs also stressed the importance of adequate medical documentation, as they collaborate more with adequate record-keeping practices. Patients emphasised the importance of including the outcomes of the shared decision making in the transfer between the physiotherapists and GPs.

For the transition from the third to fourth stage, the GP stated that it could help to have some referral templates for care that requires an official referral. Moreover, patients emphasised that care should be coordinated by a healthcare professional, so they should not have multiple visits with different healthcare professionals for the same goal.

> P-4: *"I also think that if you are in such a situation, you probably also need home care, so from multiple domains healthcare professionals will be approaching you. So that might be a lot of the same from different directions, that does not seem effective to me. [. . .] That is of course a waste of energy and money."* (Q7)

## 2. Organisation & execution

In this theme, we present facilitators and barriers to the implementation in subthemes related to the processes within the TFCP, as provided by patients and healthcare professionals.

**2.1. Case finding at the ED.** Starting in the first stage, the tasks within the TFCP must be clear for ED nurses and ED doctors. They stated that carrying out tasks related to an implementation study, e.g., an informed consent procedure, would be a barrier for the implementation and should be one of the researchers' responsibilities.

> EDN-1: *"At the ED, we like practical agreements, so the shorter, the more concise, the clearer, the better."* (Q8)

> EDN-1: *"However, if it is truly about informing about the study and informed consent procedure than I wonder if that is the task of the ED nurse. Screening and selecting for the care, sure."* (Q9)

ED nurses and ED doctors agreed that triage could be the first moment to detect TFCP patients. Therefore, it is important that both ED nurses and ED doctors are familiar with the TFCP. The triage nurse could make a quick remark to the ED nurse regarding the TFCP, aiming to initiate action and enhance awareness that the patient may be a potential TFCP patient. Since not all of the TFCP's target population criteria are ascertainable at the onset of an ED visit, e.g., no surgery or long hospitalisation, the ED nurse has the opportunity to consider approaching the patient later in the process. Ideally, patients want to be approached between diagnosis and sometime before discharge. During this time, they have time to process the information and ask questions to the ED doctors or ED nurses. The materials to approach patients, e.g., pamphlets, have to be easily accessible for ED staff, and the approach should not take too long. In case of a language barrier or confused patient, the materials must be available for friends or family of the patient. The healthcare professionals stated that informing patients of the TFCP should not take longer than five minutes. However, the ED healthcare professionals did state that the identification of patients will have less priority during crowded hours.

> EDD: *"We should try to create a feeling that despite hectic times at the ED, the patient could deteriorate over time if we don't start the TFCP."* (Q10)

In accordance with the time aspect, the TFCP's target population criteria should be short and easily interpretable for the ED staff. For example, we discussed the following criterion with the ED staff which was deemed too vague: "Patient's life expectation has to be longer than one year". Another criterion is a Clinical Frailty Score of 4 to 6. This was not necessarily deemed easy but useful enough. Especially when working with other specialisations at the ED. The ED doctor is not always the primary caregiver, which might be a barrier to implementation, as other specialisations are less familiar with the TFCP. The Clinical Frailty Score criteria felt as a strong argument by the ED nurses to convince other specialisations to include the patient. Some nurses are more specialised in geriatric care and could act as champions for the TFCP towards other specialisations in the ED. Champions are individuals who enthusiastically promote and facilitate implementation of innovations [39].

Healthcare professionals indicated that there are already some standard workflows that could help the implementation. For example, ED nurses already register some aspects of a patient's fall risk in the electronic patient record. A pop-up or reminder in the electronic patient record could help remember to think about the TFCP.

> EDN-1: *"My preference is that we simply implement this in our electronic patient record in which we already screen for falls and fall risk. If we could link these questions to a reminder of*

*the TFCP, you can't really ignore it. But then it has to be built in, that is where we will face other challenges." (Q11)*

The healthcare professionals stated that the functions with regard to pop-ups and transmural communication in the electronic patient record system should be explored to enhance case finding and initiating the TFCP. Moreover, the electronic patient system can be used by colleagues such as the coordinating nurse to help recognise potential TFCP patients as they keep an overview.

**2.2. Fall risk assessment domains.** Some of the physiotherapists in the focus groups also participated in the pilot study. They stated that the fall risk assessment took too long but does give a comprehensive overview of the relevant fall-related risk factors. Patients indicated that the fall risk assessment should include at least: fall history, mobility, balance, strength, activity level, walking pattern, home environment, walking aids, shoes, necessity of home care, nutritional status, fear (of falling), vision and cognition.

Patients indicated the need for more comprehensive care coordination, as they preferred not to discuss the same items of the fall risk assessment with multiple healthcare professionals.

**2.3. Engagement and practical barriers of tailored care.** Patients want a holistic approach in which care is tailored to their needs and level. Participation is facilitated when the intervention is experienced as positive and fun. The intervention can be perceived as a pleasant disruption of the week to get patients moving. They attributed this to the social aspect of an intervention, especially group interventions, as patients enjoy the company. A group can offer perspective to continue social activities after the intervention period. However, not all patients find it easy to make new contacts, indicating a possible barrier. Hesitant patients could be stimulated by offering try-out sessions. Patient 010 compared it to the test session she had for a daytime activities centre. Ten years later, she still enjoys going to the centre. Another factor that can contribute to the pleasantness of the intervention can be the results patients experience over time. Patients do state that the lack of results can be demotivating as well. Furthermore, there is a call for long-term care planning after the initial intervention period.

P-6: *"I think that participating in the indicated care is the first phase. The next phase is when patients continue with each other. . . . I think that a patient who needs to lose weight or who is unstable could benefit from behavioural change, not just a short-term intervention." (Q12)*

More practical barriers of the intervention, presented by patients, can be related to the accessibility of the intervention. A large travel distance to the intervention will make it hard for patients to get there. The interviewed patients indicated that an intervention within their own neighbourhood would be optimal. Another practical barrier can be the costs of an intervention. If patients have to pay the full costs, participation can be withheld. However, once patients have paid, it can be a facilitator to complete the intervention since they already paid for it. Some patients stated that a (partial) reimbursement by insurance companies could motivate but is also something they see as necessary for the importance of prevention.

**2.4. Finances.** A possible barrier presented by patients is the absence of healthcare insurance coverage for fall risk assessment and fall prevention interventions. In 2023, fall risk assessments and physiotherapy are not covered by basic insurance in the Netherlands. However, the participating patients stated that they were willing to pay for the fall risk assessment, but insurance coverage would be a facilitator. Ten hours of occupational therapy is covered by basic health insurance and could provide a possibility for patients without additional insurance. There are promising national developments regarding fall prevention, as fall risk assessments will be covered for frail older adults in 2024.

WMC: "*Does it matter if the TFCP is covered by insurance?*"

P-5: "*I believe it does. . . . I just think it is bad of them (insurance companies), truly. You can include that too. I just think that if they want people to remain mobile, then physical therapy should receive more attention, especially for older adults because many can't afford that.*" *(Q13)*

**2.5. Time.**   In all stages of TFCP and for all involved people, time is a recurring theme. Patients want to indicate a preferred time for the fall risk assessment. Furthermore, the indicated care can disrupt a busy standard week. Patients need to make time for the interventions, and not all will be willing to change their schedules. At the ED, recognition and inclusion of patients should be time efficient for ED doctors and ED nurses.

EDN-1: "*Patient recognition and inclusion at the ED has to be quick and effective. It should not cost too much time; otherwise, people won't do it.*"

MVB: "*Within what time frame should it be possible?*"

EDN-1: "*Within a few minutes seems right to me*"

EDN-2: "*Yes, a maximum of five minutes.*" *(Q14)*

The time investment of the fall risk assessment can be a barrier for the physiotherapists to complete their part in the TFCP. Attempting to contact the GP or PN for the transferal can be time consuming for the physiotherapists as well. On their end, the GPs and PNs are generally quite busy, thus having limited time to invest in the TFCP.

**2.6. Responsibilities.**   One of the central discussions with the healthcare professionals was about who should take the lead in the TFCP. In general consensus, the GP should have the management of the overall picture. However, as the GP might not yet be up to date during stage 2, the physiotherapists have the lead at that stage. After the transfer of results, the GP and PN should take the lead. Patients stated they had no problem with the PN taking the lead due to the GP's high workload.

WMC: "*Would it help if the GP would contact you after receiving the transferral from the physiotherapists?*"

P-2: "*Yes, but you also don't want to burden doctors, especially during the pandemic. I often interact more with the PN than the GPs themselves.*" *(Q15)*

A specific barrier regarding responsibilities occurred. We explored the possibilities of a medication check at the ED. In this check, ED doctors would control fall risk-increasing drugs and include advice to the GP in the discharge letter. An important barrier to changing drugs according to the healthcare professionals is that patients often receive drugs from multiple physicians. Doctors are not always comfortable with changing the drugs that are prescribed by other physicians. The GPs and PNs indicated that the ED-doctor's advice could be used in regular meetings with pharmacists.

## 3. Personal factors

In this last theme, we revealed personal factors affecting the actions of both patients and healthcare professionals within the TFCP.

**3.1. Emotions and behaviour.**   During their ED visit, patients reported being focused on short-term outcomes. They are eager to learn their diagnosis and prognosis. During their ED

stay, patients did not feel an urge to prevent another fall or ED visit. Patients describe their fall as an accident and a one-time occurrence.

> WMC: "*Did you feel an urge for action to prevent another fall while you were at the ED?*"
>
> P-4: "*No, it was just a dumb accident. . . . I would like to write the municipality to fix all those uneven paving stones.*" (Q16)
>
> P-10: "*I don't think it will help, it will solve itself eventually. I will just hold on tight (laughs), I don't believe it is necessary.*" (Q17)

There was a variety of responses to the question of whether patients would have liked to participate in the TFCP. Some patients stated that they would not have wanted it since they had other worries or they felt like they did not need it. Others stated that they would have loved it for more detailed attention to their healthcare problems. It felt comforting that a variety of healthcare professionals could easily be contacted for help with the TFCP. However, almost none of the participants felt that they were part of the target population.

As stated before, patients stated that they can be quite overwhelmed at the ED. Some experience pain, shock, loneliness or fear. Others are angry at the occurrence of the fall. Upon returning home, patients can be uncomfortable due to the pain and the list of items they have to arrange, e.g., home care and groceries. As patients are (temporarily) more dependent, they reported to act more carefully and become more anxious. In some cases, patients experienced difficulties sleeping or are gloomy. In particular, older adults who live alone dislike coming home to an empty home. The help of family and friends can help during these times. Other older adults indicated that they were relieved and reassured to be home again.

**3.2. Knowledge.** The researchers experienced that not all patients recognised the importance of fall prevention. Some stated that this was due to the mechanism of their fall or because they viewed it as a one-time only incident. During certain interactions between researchers and patients, it became apparent that the target population had limited knowledge of the concept and content of fall prevention. Some older adults believed that improvement in function is no longer possible due to their age.

> P-3: "*I think I have given up, I have accepted it. I think the same as the doctors says: "Think about your age". Then I think, I don't know if I can make progress.*"
>
> WMC: "*Why do you feel that you can't make progress?*"
>
> P-3: "*I feel like a wreck because walking is difficult and I'm always tired. I don't feel like I can make progress, at my age. [. . .] I think I gave up and that will stay until I die, and I have accepted that even though it is hard. I'm not going to participate in different activities like walking in the woods. I have become lazy and inactive, and I just accepted it.*" (Q18)

Due to these beliefs and gaps in knowledge, it was difficult for patients to view themselves as the target population of the TFCP, making it an important barrier for participation. Not only patients but also healthcare professionals from the ED and primary care had trouble understanding the content of fall prevention. The healthcare professionals stated that presence of fall prevention in the national political coalition agreement and in quality indicators by the Ministry of Health, Welfare and Sport will facilitate the acquisition of knowledge. Furthermore, the ED indicated that educational meetings with ED nurses and ED doctors can improve awareness and knowledge on this subject. The presence of ED doctors and ED nurses with a strong interest in geriatric care and fall prevention will generally help with the

awareness of TFCP among colleagues. The same principle applies to the physiotherapists, GPs and PNs.

The physiotherapists indicated that additional education is necessary to perform the fall risk assessment, as the measurement instrument takes some getting used to. To advise the indicated care to the patient, it is important that the physiotherapist has broad knowledge on the possibilities related to specific fall risks. They indicated that geriatric physiotherapists are more capable of this overview compared to general physiotherapists. Including physiotherapists specialised in geriatric care in the TFCP would be a facilitator. This knowledge is important in the shared decision-making with the patient, so the patient can make considered choices. One of the patients stated that healthcare professionals might benefit from the existence of a knowledge centre.

It was also stated that by healthcare professionals that not all GPs are always familiar with the multifactorial aspect of fall risks and fall risk assessment. This can be a barrier in the communication and transfer between the physiotherapists and GPs.

**3.3. Motivation and activation.** The patients described that the attitude, beliefs and knowledge of patients can be important facilitators or barriers. Some patients are perseverant and want to recover as quickly as possible, while others could be described as "care avoiders" or do not feel addressed. In some cases, patients are too weak or too ashamed of their fall and therefore refuse to seek help. The patient's previous experiences with an intervention could be facilitators or barriers depending on the nature of the experience. Patients who see their health as a priority will be more eager to participate in the indicated care. Those who are slightly more hesitant could be motivated if they want to remain healthy, live independently and not be a burden for their spouse.

> P-3: *"I don't want to use a cane for the rest of my life. . . . I would also like to do it to stop my partner from having to act as caregiver." (Q19)*

> P-5: *"You want to grow old pleasantly don't you? If you become dependent, that is horrible, you want to postpone that as long as possible." (Q20)*

Patients felt like they could be motivated with examples of other older adults in which progress was achieved. The previously stated risk explanation of the current situation by the therapists could also stimulate participation. This is especially true if the therapist is perceived as competent, kind and interested. However, it can be difficult to accept that interventions are necessary for the current situation. Contact from the GP or PN on the results of the fall risk analysis could help this process.

> WMC: "*Would you like for the GP to contact you after the fall risk assessment, to discuss the next steps?*"

> P-5: "*Yes, but I do mind that the GP will have extra work.*"

> WMC: "*Would the contact encourage you to actually take the next steps?*"

> P-5: "*For sure, I think patients would be more motivated if the GP emphasises that they should do it and that interventions will be beneficial for them." (Q21)*

Patients stated that once they would have accepted the necessity and experienced the progress, they could become more engaged in their treatment plan. At the start of the treatment plan, the previously mentioned test sessions can decrease the barrier to participation. Moreover, the patients stated that interventions that are perceived as positive, fun and socially

important will a facilitating effect. The healthcare professionals associated with the indicated care can be important facilitators or barriers for patients. Similar to the therapist from the frailty assessment, the patients would like the healthcare professional to be kind, interested and competent. Verbal and physical accessibility of a healthcare professional is important.

Healthcare professionals stated that they can be motivated by using a reward system and news letters to keep them informed about the process. Other relevant factors that were previously mentioned are interest in geriatric care, education and external factors such as quality indicators.

> P-5: *"I want to stay for a while. . . I know some older adults who have a little ache and therefore remain in their chairs. That is the beginning of the end, you have to stay active, it is that simple."* (Q22)

## Discussion

In this qualitative study, we identified facilitators and barriers for the implementation of a TFCP for older adults after a fall-related ED visit from the perspectives of patients and healthcare professionals. We divided the facilitators and barriers into three key themes.

The first theme, 'communication', consists of communication between patients and healthcare professionals, interprofessional communication, and communication between patients and their families. In the second theme, 'organisation & execution', we revealed subthemes on facilitators and barriers in the processes throughout the TFCP, such as case finding at the ED, fall risk assessment, engagement and practical barriers of tailored care, time, responsibilities, and finances. In the third theme, we presented the facilitators and barriers related to personal factors of patients and healthcare professionals such as emotions and behaviour, knowledge, and motivation and activation.

Interestingly, most subthemes within each theme can act as both facilitators and barriers. For example, empathic communication by a healthcare professional acts as a facilitator, while belittling or hierarchical communication acts as a barrier. Additionally, using existing workflows and notifications in electronic patient records facilitates case finding, while time constraints due to crowded EDs act as a barrier for case finding. In terms of personal factors, patients' attitudes towards their health can both be a facilitator and a barrier. This duality becomes apparent when comparing patients who prioritise their health and those who tend to avoid care. Patients who prioritise their health are more inclined to engage in the TFCP, while the latter group is less likely to do so.

This is one of the first studies on facilitators and barriers for the implementation of a TFCP from the ED to primary care beside the pilot study by *Hepkema et al* [18]. To explain our findings, we compare our results with studies focusing on facilitators and barriers in various contexts, including transmural fall prevention after acute hospitalisation, transmural mental health care, transmural palliative care, transmural care transitions, and community-based fall prevention [40–44].

The key facilitators found in our study aligning with prior research include the use of specialised healthcare professionals in transmural care pathways, tailoring care plans to suit the needs and wishes of patients, the presence of social support in the patient's environment, and using patients' intrinsic motivation to remain independent to engage in care [18, 41–44].

For barriers, similar to previous studies, transmural collaboration presents a challenge in various settings, as a significant barrier therein is the absence of clear agreements on responsibilities between healthcare professionals [41–43]. Another barrier we found that was described

in other studies was the presence of logistical barriers. Patients and healthcare professionals across various disciplines often struggle with financial, organisational or transportation issues [18, 40–44]. Additionally, healthcare professionals in transmural care face barriers getting in contact with colleagues, in our study both between the ED and the GP and between the GP and PT [18, 40, 42, 43].

Similar to other studies we find that older adults may fail to recognise their own fall risk [18, 40, 42–44]. This aligns with the 'better for others than for me' phenomenon as described by Haines *et al* [45]. In this phenomenon, older adults may recognise the benefits of fall prevention interventions in general but refrain from participating because they believe it will not benefit them personally.

There were also discrepancies between our facilitators and barriers compared to the literature. In the study on facilitators of and barriers to transmural fall prevention after acute hospitalisation, the authors found that prolonged community engagement by healthcare professionals and healthcare institutions and relationship building with older adults mitigated barriers to participation [44]. These facilitators were not mentioned by participants in our study. Moreover, another prominent barrier the authors encountered was that patients expressed the presence of health-related issues, such as joint or chest pain, as a barrier for participating in fall prevention interventions after acute hospitalisation [44]. Patients in our study did mention that they wanted the interventions to be fitted to their level of physical ability but that is not quite the same. Another barrier from the study on transmural mental health care was that patients were unaware of the available resources that could be used to pay for care if they were not insured, which could be similar for fall prevention interventions [40].

Compared to the previously mentioned studies, we found some new facilitators. For example, the detailed instructions for explaining a TFCP to patients by ED healthcare professionals, including involvement of family, and the facilitating effect of multiple sources of information. This study captured the perspective of patients on what they found to be important items in a fall risk assessment. When implementing a TFCP, organisers should take into account the facilitating effect of a follow-up by a health care professional, such as the GP or PN, after receiving the physiotherapist's report. Moreover, patients also stated that they had no trouble with the PN coordinating the care instead of the GP. Preferences can differ between healthcare professionals and patients. It is therefore important to actively engage with both patients and healthcare professionals during the process of developing optimal implementation strategies.

Going one step further, some of our findings could be extrapolated into the notion of "interprofessional collaboration". This can be a challenging part of the TFCP as it involves different healthcare professionals from different institutions working together. Rawlinson *et al* conducted an overview of reviews on facilitators and barriers in interprofessional collaboration [46]. They found that the reviews reported the factors at the organisational and inter-individual levels. Based on their research, barriers that may be experienced during the implementation of the TFCP could be lack of time, training and clear roles. In addition, professionals may experience fears related to their professional identity and poor communication. Ineffective interprofessional communication can lead to poor patient outcomes and therefore an ineffective TFCP [47]. Tools to improve communication could act as facilitators [46]. In addition, co-location and recognition of each other's skills and contributions facilitate interprofessional collaboration. Researchers should integrate these insights when forming implementation strategies for the TFCP.

Another factor to consider is the high staff turnover in emergency departments [48]. This has implications for sustainability beyond the initial implementation period of the TFCP [49]. During the implementation of a TFCP, researchers should consider the increased need for training, loss of organisational knowledge, lack of fidelity to evidence-based practice and

possible financial stress. In forming strategies, researchers should pay more attention to the use of champions and the alignment of the TFCP with organisational goals [49].

Although the identification of the different facilitators and barriers, and the subsequent development of tailored implementation strategies, may suggest that implementation will be a smooth process, reality bears witness to the contrary [50, 51]. Researchers seeking to implement a TFCP should therefore be mindful of our findings but be prepared for unexpected challenges and remain alert to new barriers and facilitators during implementation.

## Strengths and limitations

It is important to acknowledge some limitations of this study. Older adults in the VUmc area generally have a relatively high socioeconomic status, which may influence the generalisability of the found facilitators and barriers. In lower socioeconomic areas, differing healthcare access might reveal additional or more prominent barriers that were less emphasized in this study [52]. Additionally, data saturation with the healthcare professionals may not have been reached. Due to the high workload in primary care during the COVID pandemic, we were unable to organise a focus group with all relevant healthcare professionals at the same time. The high workload also caused certain professions, such as the GP, to be underrepresented in our study. This may impact the generalisability of the results as the perspectives of a few individuals may not fully represent the entire professional group.

Our study also has several strengths. First, the integration of viewpoints from both patients and healthcare professionals allowed us to present comprehensive information on each subtheme. Another strength of this study is that the participants were representative of the TFCP's target population in the geographic area of Amsterdam UMC, location VUmc. The age range of the patients was between 66 and 90 years. None of the patients were familiar with fall prevention, which is common for older adults visiting the ED with fall-related injuries. It provides better insight into the challenges to motivate people for fall prevention interventions in the Netherlands.

The use of a constructivist paradigm has contributed to our findings as it acknowledges the subjective nature of realities and the differences in individual perspectives. By exploring these multiple perspectives, we were able to provide a more comprehensive and inclusive understanding of the themes. The interaction between researchers and participants may have impacted the results due to the radical subjectivism. In this case, interviewers' backgrounds may have influenced the results, potentially skewing them towards the perspectives of healthcare professionals rather than patients. To mitigate this bias, we pilot tested the interviews with patient representatives. The interviewers' background also led to more specific follow-up questions, such as patients' experiences with treatment. The interviewers, being actively involved in patient care, were more familiar with the barriers or facilitators that could arise during treatment. Additionally, their knowledge of the TFCP's process allowed them to explain it to the patients, resulting in higher quality discussions.

In implementation research, interview guides are frequently shaped by a conceptual framework, such as the Consolidated Framework for Implementation Research (CFIR) [53, 54]. In this study, we used the concept of the TFCP as a conceptual basis for developing our interview guide in collaboration with our consortium. Rather than relying solely on the CFIR, we aimed to gather input directly from participants regarding what they deemed most important. However, we did incorporate certain CFIR constructs, such as adaptability, available resources and access to knowledge & information. In retrospect, if we had based our interview guide more heavily on these constructs, the results may have differed. It is worth noting that these constructs are generally more focused on corporate structures or opportunities, and less on

personal factors. Our approach prioritised the perspectives of patients and healthcare professionals, resulting in a strong personal voice.

## Conclusion

It is crucial to consider facilitators and barriers from the perspectives of both patients and healthcare professionals for the implementation of a TFCP in communication, organisation & execution, and personal factors. The results of this study provide useful information to design tailored implementation strategies. The implementation of a TFCP will guide patients towards fall preventive interventions, ultimately reducing the risk of recurrent falls and fall-related ED visits. The found themes are relevant to take into account when designing implementation strategies in all settings, but future research is needed to explore how additional or different barriers may emerge in lower socioeconomic areas. Furthermore, future research should evaluate the effectiveness of tailored implementation strategies for the implementation of a TFCP.

We would also like to thank Marlou van Beneden (MvB) for her help as focus group leader.

## Supporting information

**S1 Appendix. COREQ criteria.**
(PDF)

**S2 Appendix. Patient interview guide.**
(DOCX)

**S3 Appendix. Focus group 1 interview guide.**
(DOCX)

**S4 Appendix. Focus group 2 interview guide.**
(DOCX)

**S5 Appendix. Original quotes in Dutch.**
(PDF)

**S1 Table. Patient characteristics per participant.**
(PDF)

**S2 Table. Demographic characteristics per healthcare professional.**
(PDF)

## Acknowledgments

We would like to acknowledge all members of our consortium for helping with the construction of the interview guide and discussions of the results: Osteoporose Vereniging, Zorg voor Zuid Amsterdam, GGD Amsterdam, Gemeente Amsterdam, Cliëntenraad Amsterdam UMC, Fysiotherapie Collectief Amsterdam, VeiligheidNL, Zilveren Kruis en Koninklijk Nederlands Genootschap voor Fysiotherapie (KNGF).

## Author Contributions

**Conceptualization:** W. M. Charmant, B. A. M. Snoeker, H. P. J. van Hout, E. Geleijn, C. Veenhof, P. W. B. Nanayakkara.

**Formal analysis:** W. M. Charmant, I. N. Nauta, F. Boonstra.

**Funding acquisition:** B. A. M. Snoeker, H. P. J. van Hout, E. Geleijn, C. Veenhof, P. W. B. Nanayakkara.

**Investigation:** W. M. Charmant, I. N. Nauta, F. Boonstra.

**Methodology:** W. M. Charmant, B. A. M. Snoeker, H. P. J. van Hout, E. Geleijn, C. Veenhof, P. W. B. Nanayakkara.

**Project administration:** W. M. Charmant, B. A. M. Snoeker.

**Supervision:** B. A. M. Snoeker, H. P. J. van Hout, E. Geleijn, C. Veenhof, P. W. B. Nanayakkara.

**Visualization:** W. M. Charmant.

**Writing – original draft:** W. M. Charmant.

**Writing – review & editing:** B. A. M. Snoeker, H. P. J. van Hout, E. Geleijn, C. Veenhof, P. W. B. Nanayakkara.

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
