## [Decision Letter · Decision Letter 0]

12 Dec 2023

PONE-D-23-33190Facilitators and barriers for the implementation of a transmural fall-prevention care pathway for older adults in the emergency departmentPLOS ONE

Dear Dr. Snoeker,

Thank you for submitting your manuscript to PLOS ONE. After careful consideration, we feel that it has merit but does not fully meet PLOS ONE’s publication criteria as it currently stands. Therefore, we invite you to submit a revised version of the manuscript that addresses the points raised during the review process.

1. Kindly modify your abstract by highlighting key specific findings or insights from the themes and sub-themes from your study. In addition, "the conclusion could emphasize the potential impact of these findings on fall-prevention strategies in emergency departments.".2.  In your Methods section kindly justify your sample size selection, including inclusion and exclusion criteria. And also elaborate a bit further on the demographic characteristics of your study participants. Kindly elaborate on your qualitative methods including who conducted the interviews, pilot testing, and why you chose individual interviews/ focus groups and who participated in or conducted the focus group discussions, and who transcribed the data, etc.3. Kindly submit a copy of your Interview guide as a supplementary file to accompany your revised manuscript.3. In your results section kindly add more quotations where necessary to better illustrate your findings. Kindly reorganize your themes and sub-themes for better clarity and depth. Kindly clarify confusing definitions or statements such as 'the shared decision-making conversation'  and "implementation strategies". Also clarify 'barriers' and 'facilitators' in your study.4. "The discussion could benefit from a more extensive comparison with existing literature, discussing how these findings align or differ from previous studies". Consider including 'a discussion on the use of a constructivist paradigm in this study.' Also, include a clear of statement of Strengths and Limitations of your study after the Discussion section.5. Since this is is a qualitative study, please submit a fully completed COREQ or SRQR Checklist as a supplementary file with your revised manuscript. And in your Methods include a statement that the study was completed in accordance with either of these guidelines.6. Kindly address all other issues raised and annotated by Reviewer 2, including providing appropriate citations where necessary.

We look forward to receiving your revised manuscript.

Kind regards,

Sylvester Chidi Chima, M.D., L.L.M.

Academic Editor

PLOS ONE

Journal Requirements:

Reviewers' comments:

Reviewer's Responses to Questions

**Comments to the Author**

1. Is the manuscript technically sound, and do the data support the conclusions?

Reviewer #1: Yes

Reviewer #2: Partly

2. Has the statistical analysis been performed appropriately and rigorously? 

Reviewer #1: Yes

Reviewer #2: N/A

3. Have the authors made all data underlying the findings in their manuscript fully available?

Reviewer #1: Yes

Reviewer #2: Yes

4. Is the manuscript presented in an intelligible fashion and written in standard English?

Reviewer #1: Yes

Reviewer #2: Yes

5. Review Comments to the Author

Reviewer #1: It was a pleasure reviewing your work. Overall, the paper seems well-structured and informative, with a clear focus on a significant healthcare issue. However, would benefit from some minor edits. See below:

Specific comments

Abstract

The abstract provides a clear overview of the study, its methods, and the main findings. It succinctly encapsulates the study's purpose, methodology, and conclusions.

While your abstract mentions the main themes, it could benefit from briefly highlighting key specific findings or insights from these themes. Also, the conclusion could emphasize the potential impact of these findings on fall-prevention strategies in emergency departments.

Methods

Concerning your sample size justification, while the number of participants is mentioned, the rationale behind choosing ten older adults and thirteen healthcare professionals could be elaborated for clarity.

Also, more information on the diversity of the participant group (e.g., demographics, professional backgrounds) could enhance the understanding of the study's applicability.

Results

Kudos on this section. The results were well-organized, providing a structured overview of findings.

However, more integration of direct quotes or specific examples from the interviews and focus groups could enrich the narrative and support the findings.

Discussion

The discussion could benefit from a more extensive comparison with existing literature, discussing how these findings align or differ from previous studies.

What were the limitations in your study.

Reviewer #2: Please see attached file.

6. PLOS authors have the option to publish the peer review history of their article (what does this mean?). If published, this will include your full peer review and any attached files.

Reviewer #1: **Yes: **Udoka Okpalauwaekwe

Reviewer #2: No

---

## [Author Response · Author response to Decision Letter 0]

5 Feb 2024

Dear Editor,

We have been asked to make our data publically available. However, a public deposition would breach compliance with the protocol approved by our research ethics board and the terms stipulated in the informed consent agreement with our participants. In our approved protocol and participant informed consent, we agreed that interview data would only be shared upon request of individual researchers. Notably, 21 out of 23 participants have agreed for their data to be shared upon request with researchers outside of Amsterdam UMC. 

We believe that these ethical and legal reasons are sufficient for an exemption request.

To complete the statement we provided the chair of the research ethics board that approved the study:

Prof. Dr. J.A.M. van der Post

metc@amsterdamumc.nl

Kind regards, on behalf of all co-authors,

Wilmar Charmant, MSc

Please see "Response to Reviewers" file for a better overview of our responses. Tables disappear in this box. 

General comments

1. Kindly modify your abstract by highlighting key specific findings or insights from the themes and sub-themes from your study. In addition, "the conclusion could emphasize the potential impact of these findings on fall-prevention strategies in emergency departments.".

Authors response: We have specified the key themes and the specific finding in the result section of the abstract (Line 37-47). Additionally, we have added the potential impact to the abstract’s conclusion (Line 51-52). 

2. In your Methods section kindly justify your sample size selection, including inclusion and exclusion criteria. And also elaborate a bit further on the demographic characteristics of your study participants. Kindly elaborate on your qualitative methods including who conducted the interviews, pilot testing, and why you chose individual interviews/ focus groups and who participated in or conducted the focus group discussions, and who transcribed the data, etc.

Authors response: We added the description of sample size, inclusion- and exclusion criteria into “Participant recruitment and sampling” (Line 109-128). Demographic characteristics of participants are described in the “Results” section (Line 225-237). In “Data collection”, we describe who conducted the interviews/focus groups, pilot testing and who transcribed the data (Line 160-184). 

3. Kindly submit a copy of your Interview guide as a supplementary file to accompany your revised manuscript.

Authors response: We have added the interview guides to the supplementary files (S2, S3 and S4). 

4. In your results section kindly add more quotations where necessary to better illustrate your findings. Kindly reorganize your themes and sub-themes for better clarity and depth. Kindly clarify confusing definitions or statements such as 'the shared decision-making conversation' and "implementation strategies". Also clarify 'barriers' and 'facilitators' in your study.

Authors response: We have added more quotations in the results in sections that the reviewers indicated (Lines 344-347, 360-365, 385-388, 421-422 and 440-443). Moreover, we have defined implementation strategies in the background section (Lines 82-83) and clarified “shared decision-making” by removing “conversation” and adding a reference (Line 150-151). At the start of the results section we have added a sentence clarifying why factors are portrayed as either facilitators or barriers (Line 248-250). We have discussed reorganisation of the themes, especially 2b) fall risk assessment content. While respecting the reviewer’s suggestion, we believe that the theme should stand on its own as it discusses a standalone part of the TFCP. We did change the title of 2b to fall risk assessment domains, similar to the world guidelines for fall prevention and management for older adults. 

4. "The discussion could benefit from a more extensive comparison with existing literature, discussing how these findings align or differ from previous studies". Consider including 'a discussion on the use of a constructivist paradigm in this study.' Also, include a clear of statement of Strengths and Limitations of your study after the Discussion section.

Authors response: We have extended our discussion with a more in depth continuation on interprofessional collaboration, high staff turnover in the ED, and the reality of implementation (Line 731-754). Additionally, we included a paragraph on the influence of a constructivist paradigm and the lack of framework in the development of the interview guide (Line 777-799). A paragraph of Strengths and Limitations is included after the Discussion (Line 755-799). 

5. Since this is a qualitative study, please submit a fully completed COREQ or SRQR Checklist as a supplementary file with your revised manuscript. And in your Methods include a statement that the study was completed in accordance with either of these guidelines.

Authors response: In S1, a fully completed COREQ checklist is present. It is also mentioned in the first paragraph of Methods (Line 96-97). 

6. Kindly address all other issues raised and annotated by Reviewer 2, including providing appropriate citations where necessary.

Journal Requirements:

Authors response: Wilmar Charmant’s ORCID ID is registered in the Editorial Manager

Reviewer 1 comments to the manuscript: Facilitators and barriers for the implementation of a transmural fall-prevention care pathway for older adults in the emergency department.

Manuscript #: PONE-D-23-33190

Title: Facilitators and barriers for the implementation of a transmural fall-prevention care pathway for older adults in the emergency department

Authors: Willem Maarten Charmant; Barbara Snoeker; Hein van Hout; Ilse Nauta; Femke Boonstra; Edwin Geleijn; Cindy Veenhof; Prabath Nanayakkara

Article type: Research Article

Comments Possible changes Line Author’s response Adjusted lines in the unmarked version

It was a pleasure reviewing your work. Overall, the paper seems well-structured and informative, with a clear focus on a significant healthcare issue. 

However, would benefit from some minor edits. See below Thank you for your compliments regarding our work.

The abstract provides a clear overview of the study, its methods, and the main findings. It succinctly encapsulates the study's purpose, methodology, and conclusions.

While your abstract mentions the main themes, it could benefit from briefly highlighting key specific findings or insights from these themes. Also, the conclusion could emphasize the potential impact of these findings on fall-prevention strategies in emergency departments. Abstract We have specified the key themes and the specific finding in the result section of the abstract. Additionally, we have added the potential impact on EDs and primary care to the abstract’s conclusion. (Line 37-47). 

(Line 51-52).

Concerning your sample size justification, while the number of participants is mentioned, the rationale behind choosing ten older adults and thirteen healthcare professionals could be elaborated for clarity.

Also, more information on the diversity of the participant group (e.g., demographics, professional backgrounds) could enhance the understanding of the study's applicability.

 Methods We based our sample size on data saturation. Therefore, sample size was not determined prior to the interviews. Even though, after seven patient interviews, we felt that we achieved data saturation we continued up to ten interviews. We did this to make sure we achieved data saturation. For the healthcare professionals, we based our sample on the inclusion of representatives from each domain: ED, physiotherapy and GPs. 

The demographics of the participants are included in Table 2 in the Results. 

 Line 202-204

Kudos on this section. The results were well-organized, providing a structured overview of findings.

However, more integration of direct quotes or specific examples from the interviews and focus groups could enrich the narrative and support the findings.

 Results Thank you for your compliments. We have added more quotes to the results section. (Lines 344-347, 360-365, 385-388, 421-422 and 440-443)

The discussion could benefit from a more extensive comparison with existing literature, discussing how these findings align or differ from previous studies.

What were the limitations in your study. Discussion We have elaborated our discussion with a more in depth continuation on interprofessional collaboration, high staff turnover in the ED, and the reality of implementation. Additionally, we included a paragraph on the influence of a constructivist paradigm and the lack of framework in the development of the interview guide. A paragraph of Strengths and Limitations is found after the Discussion. Our main limitations are described in strengths and limitations. (Line 731-754). 

(Line 777-799).

Reviewer 2 comments to the manuscript: Facilitators and barriers for the implementation of a transmural fall-prevention care pathway for older adults in the emergency department.

Manuscript #: PONE-D-23-33190

Title: Facilitators and barriers for the implementation of a transmural fall-prevention care pathway for older adults in the emergency department

Authors: Willem Maarten Charmant; Barbara Snoeker; Hein van Hout; Ilse Nauta; Femke Boonstra; Edwin Geleijn; Cindy Veenhof; Prabath Nanayakkara

Article type: Research Article

Comments Possible changes Line Author’s response Adjusted lines

Thank you for the opportunity to review this manuscript. It focuses on a highly clinically relevant issue, namely, fall patients. In terms of the manuscript's quality regarding qualitative studies, several revisions are needed. See comments below. Thank you for your compliments regarding our study. 

On line 81, it is written, 'to develop implementation strategies,' but often, we choose existing described strategies, as seen in, for example, Powell et al., 2012, 2015, rather than developing our own implementation strategies. So, perhaps it is more accurate to write 'to select and/or develop...' Line 81 Adjusted to “select and develop”. 

Waltz TJ, Powell BJ, Fernández ME, Abadie B, Damschroder LJ. Choosing implementation strategies to address contextual barriers: diversity in recommendations and future directions. Implement Sci. 2019;14(1):42. Line 83-86

The section on the epistemological standpoint lacks proper references.

 Line 97-101 Reference 21 described the epistemology “radical subjectivism”. For further clarification, we now added the paper by Mark Olssen (REF 26) on the epistemology of constructivism. This does not use the same term “radical subjectivism” but describes the same concept. Line 100-106

In line 121 is written ‘The healthcare professionals were

122 familiar with care for older adults after a fall’. What does familiar means in the context of recruitment of health care professionals?

Were other selected inclusion criteria used, such as general professional experience and experience with the specialty? Why or why not?"

 Line 121-122 Familiar means that the healthcare professionals regularly treated older adults within their profession. We changed the sentence for more clarification. 

Physio- and occupational therapists were experienced with fall risk assessments as they were affiliated with the pilot study. We have added thist to “Participant recruitment and sampling” Line 115-116

Line 116-118

Could you elaborate a bit more on the choices you made regarding settings? Why were interviews conducted in the patients' homes and not at the hospital? Why did you choose to categorize healthcare professionals into primary and secondary staff when your objective was to develop a cross-sectoral pathway? Line 126-133 Patient interviews were conducted at their home so they would be more comfortable. Furthermore, by offering to come to them, patients would be more likely to participate as they would not have to travel themselves. We described our choices more detailed now in Setting.

We tried to get all healthcare professionals into one focus group but logistical barriers due to the COVID pandemic made this impossible. Therefore we divided it into two groups. This is a limitation of the study and is described in the paragraph strengths and limitations of the discussion section. Line 133-134

Line 761-768

On line 136, it is written ‘We designed the concept of TFCP for older adults after a fall-related ED visit’. I wonder who 'we' are. Is it the research group behind the study? What is the basis for developing this program? Is it based on experiences, evidence-based literature, inputs from users etc.? How was this intervention developed?" Line 236-145 The concept TFCP was designed by the research group based on prior experiences from the pilot study. In the pilot study they designed the care pathway together with a consortium of healthcare professionals and patient representatives. Based on lessons learned, the new concept TFCP was designed, together with the same consortium. 

Hepkema BW, Köster L, Geleijn E, E VDE, Tahir L, Osté J, et al. Feasibility of a new multifactorial fall prevention assessment and personalized intervention among older people recently discharged from the emergency department. PLoS One. 2022;17(6):e0268682.

To clarify, we have added this to the manuscript in “The transmural fall-prevention care pathway”. Line 144-145

In line 152 it is mentioned ‘were asked for their demographic characteristics’. It's a very broad and generalized statement. Could you elaborate on which demographic data and why these? Line 152 We added a new sentence which describes the specific demographic characteristics. These were chosen to give the reader a general impression of the participants. The more elaborate demographics in the appendix can be used to match the characteristics with quoted participants. Line 161-168

Line 155 was the interview guide develop in collaboration with the entire research group or selected researchers. Who and why? Line 155 The interview guide was developed with the research group and consortium to ensure we gathered the facilitators and barriers on multiple domains. 

We have added this to the text. Line 169-171

On line 156 it is written ‘After and during the discussion of

157 these themes, we reserved time for new input’. What does ‘we reserved time for new input’ means? Who was supposed to provide new inputs? When? Was it part of a pilot test of your interview guide or...? Line 156 This meant that we reserved time for participants to bring up new subjects that we did not define prior. This was already a part of the pilot interview guide. We adjusted the sentence for clarification. 

 Line 171-172

Why are WMC and IN conducting individual interviews? Who are they? Similarly, why are WMC, FB, MvB, and EG conducting the focus group interviews? Did all four participate in each focus group interview, or were they divided? How many patients in total participated, and how was this number chosen? The same goes for healthcare professionals? Line 159-162 WMC and IN conducted the individual interviews as they had the expertise and availability to conduct 

---

## [Decision Letter · Decision Letter 1]

5 Nov 2024

PONE-D-23-33190R1Facilitators and barriers for the implementation of a transmural fall-prevention care pathway for older adults in the emergency departmentPLOS ONE

Dear Dr. Charmant,

Thank you for submitting your manuscript to PLOS ONE. After careful consideration, we feel that it has merit but does not fully meet PLOS ONE’s publication criteria as it currently stands. Therefore, we invite you to submit a revised version of the manuscript that addresses the points raised during the review process.

Kindly address all issues raised by Reviewer 3 as summarized in the peer reviewers comments below.

We look forward to receiving your revised manuscript.

Kind regards,

Sylvester Chidi Chima, M.D., L.L.M.

Academic Editor

PLOS ONE

Journal Requirements:

Reviewers' comments:

Reviewer's Responses to Questions

**Comments to the Author**

1. If the authors have adequately addressed your comments raised in a previous round of review and you feel that this manuscript is now acceptable for publication, you may indicate that here to bypass the “Comments to the Author” section, enter your conflict of interest statement in the “Confidential to Editor” section, and submit your "Accept" recommendation.

Reviewer #1: All comments have been addressed

Reviewer #3: All comments have been addressed

2. Is the manuscript technically sound, and do the data support the conclusions?

Reviewer #1: Yes

Reviewer #3: Yes

3. Has the statistical analysis been performed appropriately and rigorously? 

Reviewer #1: Yes

Reviewer #3: N/A

4. Have the authors made all data underlying the findings in their manuscript fully available?

Reviewer #1: No

Reviewer #3: Yes

5. Is the manuscript presented in an intelligible fashion and written in standard English?

Reviewer #1: Yes

Reviewer #3: Yes

6. Review Comments to the Author

Reviewer #1: Thanks for the opportunity tot review your work again. Thanks for your careful attention to reviewer comments. I appreciate the improved quality and delivery of your work. Looking forward to reading your work in the future.

Sincerely

Reviewer #3: Thank you very much for allowing me to review your manuscript. Congratulations on your work, and I am sending some minor revisions that I hope will further enhance the message you wish to convey to readers:

Sample Size Justification: While the sample size is mentioned, it would be beneficial to clarify why ten older adults and thirteen healthcare professionals were selected. Expanding the description of participant diversity, such as demographic and professional backgrounds, could enrich the study's applicability.

Methods: Regarding the interviews conducted in patients' homes, the description of the field notes could be more specific. Providing additional details on how these notes were used and the basis for their utility would be helpful.

Study Limitations: The text acknowledges that data saturation was not reached in the focus groups, representing a limitation in interpreting results. It also mentions that workload and staffing changes restricted the participation of some relevant professionals. Emphasizing how this limitation affects the generalizability of the findings would be valuable.

Impact of Socioeconomic Environment: Considering that the study was conducted in an area with a high socioeconomic level, it is suggested that facilitators and barriers may vary in settings with different socioeconomic conditions. Indicating how this limits the findings' applicability to other contexts could strengthen the limitations section.

Conclusion: The current conclusion provides a good overview, but reinforcing the recommendation to implement intervention strategies in contexts with similar characteristics would be useful.

I hope my recommendations will help improve the article's clarity and precision and strengthen the practical applicability of the findings for other researchers and professionals in the field of fall prevention among older adults in hospital settings.

7. PLOS authors have the option to publish the peer review history of their article (what does this mean?). If published, this will include your full peer review and any attached files.

Reviewer #1: **Yes: **Udoka Okpalauwaekwe

Reviewer #3: No

---

## [Author Response · Author response to Decision Letter 1]

6 Nov 2024

Point to point response can be found in the attached Response to reviewer document

---

## [Decision Letter · Decision Letter 2]

19 Nov 2024

Facilitators and barriers for the implementation of a transmural fall-prevention care pathway for older adults in the emergency department

PONE-D-23-33190R2

Dear Dr. Charmant,

We’re pleased to inform you that your manuscript has been judged scientifically suitable for publication and will be formally accepted for publication once it meets all outstanding technical requirements.

Kind regards,

Sylvester Chidi Chima, M.D., L.L.M, LLD.

Academic Editor

PLOS ONE

Reviewers' comments:

Reviewer's Responses to Questions

**Comments to the Author**

1. If the authors have adequately addressed your comments raised in a previous round of review and you feel that this manuscript is now acceptable for publication, you may indicate that here to bypass the “Comments to the Author” section, enter your conflict of interest statement in the “Confidential to Editor” section, and submit your "Accept" recommendation.

Reviewer #3: All comments have been addressed

2. Is the manuscript technically sound, and do the data support the conclusions?

Reviewer #3: Yes

3. Has the statistical analysis been performed appropriately and rigorously? 

Reviewer #3: Yes

4. Have the authors made all data underlying the findings in their manuscript fully available?

Reviewer #3: No

5. Is the manuscript presented in an intelligible fashion and written in standard English?

Reviewer #3: Yes

6. Review Comments to the Author

Reviewer #3: Dear Authors,

Congratulations on the excellent work presented in your article. I would like to express my appreciation for considering my suggestions and insights in your research. Your commitment to advancing knowledge in this area is truly commendable, and I believe your work will make a significant impact.

Wishing you all the best in your future endeavors, and I look forward to seeing the continued progression of your valuable research.

Warm regards.

7. PLOS authors have the option to publish the peer review history of their article (what does this mean?). If published, this will include your full peer review and any attached files.

Reviewer #3: No

---

## [Editor Report · Acceptance letter]

16 Dec 2024

PONE-D-23-33190R2 

PLOS ONE

Dear Dr. Charmant, 

I'm pleased to inform you that your manuscript has been deemed suitable for publication in PLOS ONE. Congratulations! Your manuscript is now being handed over to our production team.

Kind regards, 

on behalf of

Professor Sylvester Chidi Chima 

Academic Editor

PLOS ONE